# The longitudinal association of stressful life events with depression remission among SHARP trial participants with depression and hypertension or diabetes in Malawi

**Kelsey R. Landrum** [1]*, **Bradley N. Gaynes**[1,2], **Harriet Akello**[3], **Jullita Kenala Malava**[3], **Josée M. Dussault**[1], **Mina C. Hosseinipour**[4], **Michael Udedi**[5], **Jones Masiye**[5], **Chifundo C. Zimba**[3], **Brian W. Pence**[1]

1 Department of Epidemiology, Gillings School of Global Public Health, University of North Carolina at Chapel Hill, Chapel Hill, North Carolina, United States of America, 2 Department of Psychiatry, University of North Carolina at Chapel Hill, Chapel Hill, North Carolina, United States of America, 3 UNC Project Malawi, Lilongwe, Malawi, 4 Department of Medicine, University of North Carolina at Chapel Hill, Chapel Hill, North Carolina, United States of America, 5 Noncommunicable Diseases and Mental Health Unit, Malawi Ministry of Health, Lilongwe, Malawi

* klandrum@email.unc.edu

**Data Availability Statement:** Data are available free of charge to researchers via NIH's data

## Abstract

Depressive disorders are leading contributors to morbidity in low- and middle-income countries and are particularly prevalent among people with non-communicable diseases (NCD). Stressful life events (SLEs) are risk factors for, and can help identify those at risk of, severe depressive illness requiring more aggressive treatment. Yet, research on the impact of SLEs on the trajectory of depressive symptoms among NCD patients indicated for depression treatment is lacking, especially in low resource settings. This study aims to estimate the longitudinal association of SLEs at baseline with depression remission achievement at three, six, and 12 months among adults with either hypertension or diabetes and comorbid depression identified as being eligible for depression treatment. Participants were recruited from 10 NCD clinics in Malawi from May 2019-December 2021. SLEs were measured by the Life Events Survey and depression remission was defined as achieving a Patient Health Questionaire-9 (PHQ-9) score <5 at follow-up. The study population (n = 737) consisted predominately of females aged 50 or higher with primary education and current employment. At baseline, participants reported a mean of 3.5 SLEs in the prior three months with 90% reporting ≥1 SLE. After adjustment, each additional SLE was associated with a lower probability of achieving depression remission at three months (cumulative incidence ratio (CIR) 0.94; 95% confidence interval: 0.90, 0.98, p = 0.002), six months (0.95; 0.92, 0.98, p = 0.002) and 12 months (0.96; 0.94, 0.99, p = 0.011). Re-expressed per 3-unit change, the probability of achieving depression remission at three, six, and 12 months was 0.82, 0.86, and 0.89 times lower per 3 SLEs (the median number of SLEs). Among NCD patients identified as eligible for depression treatment, recent SLEs at baseline were associated with lower probability of achieving depression remission at three, six, and 12 months. Findings suggest that interventions addressing SLEs during integrated NCD and depression care

repository: https://nda.nih.gov/edit_collection.html?id=2822.

**Funding:** This study, including MPIs BWP, MCH, and JM, received funding from the National Institute of Mental Health (MPIs: Brian W. Pence, PhD, Mina C. Hosseinipour, MD MPH; Jones Masiye; NIMH: U19MH113202-01). KRL received support from NIH supplement U19MH113202-04S1. JMD received support from NIH supplement U19MH113202-01S2 and NIH institutional training grant T32AI070114. The funders had no role in study design, data collection and analysis, decision to publish, or preparation of the manuscript.

**Competing interests:** The authors have declared that no competing interests exist.

interventions (e.g., teaching and practicing SLE coping strategies) may improve success of depression treatment among adult patient populations in low-resource settings and may help identify those at risk of severe and treatment resistant depression.

## Introduction

Depression prevalence is increasing globally and disproportionately in low-and middle-income countries (LMICs), further burdening health systems already faced with inequitable access to mental health care [1–3]. Approximately five percent of the global adult population, six percent of the Sub-Saharan African (SSA) population, and six percent of the general Malawian population lives with depression [3–6]. Prevalence in primary-care seeking settings in Malawi is estimated to be closer to thirty percent, as many cases are undetected in primary point of care settings [7–9].

Stressful life events (SLEs) are common, complex, and dynamic and have unique effects on individuals. These events include employment, financial, relationship, health, illness, and death-related stressors and are associated with both depression and common NCD's (such as cardio-vascular diseases) [10–14]. Experiencing SLEs and having cardiovascular disease, together, is also associated with having depression [15]. SLEs can impact depression treatment response if patients identified for depression treatment are not provided or able to practice adequate strategies to cope with and adjust to these events [16–19]. Further, recent SLEs may be able to be used to predict those at risk of severe and Treatment Resistant Depression (TRD) [20].

This study aims to estimate the longitudinal association of recent SLEs within three months of baseline interviews with depression remission at three, six, and 12 months among adults seeking hypertension and/or diabetes care at NCD clinics in Malawi who were identified as needing depression treatment. For context on this population, depression and non-communicable diseases (NCDs; e.g., cardiovascular disease (CVD) and diabetes) often occur comorbidly, and the global and Malawian mental health literature signal opportunity to integrate evidence-based interventions (EBIs) for depression with existing NCD care to achieve depression remission [6,21–23].

In Malawi, mental health care provision is severely limited. The number of mental health service providers is suboptimal, compared to the mental health needs in their patient populations, and challenges to depression care provision include limited health workforces, medical health resource (e.g., medication) gaps, and centralization of mental health services [22–24]. Mental health training, suboptimal human and health resources, and high provider workload are barriers to provision of mental health care in the national health system [23]. NCD care facilities are a possible point of intervention for depression and other mental health conditions [25]. Yet, the literature has yet to sufficiently address the impact of SLEs on depression in this population or if they can be used as a way to identify cases of depression that otherwise may have gone undetected in primary care settings.

We contribute to addressing the mental health gap in Malawi by contributing knowledge about SLE risk factors for depression among NCD patients, contributing to knowledge about possible identification points for those at risk of severe depression and TRD. Study findings can be used to improve the delivery of existing mental health interventions in integrated depression and NCD treatment settings in Malawi through addressing the impact of SLEs on depression treatment outcomes.

## Methods

### Sample and procedure

**Parent study.**  This longitudinal observational analysis is situated within the Sub-Saharan Africa Regional Partnership (SHARP) study, a clinic-randomized controlled trial comparing two sets of implementation packages to support integration of depression screening and treatment into NCD clinics across Malawi [25,26]. Clinics were randomized to an enhanced (provider leader of the intervention and a quality assurance program) or basic implementation package (provider leader of the intervention with no quality assurance program) of support strategies for integrating depression and NCD treatment services.

SHARP clinic randomized trial participants are included in this analysis. These data provide a prospectivel, a large cohort of SHARP trial participants, their SLE experiences, and their depression outcomes. The FB intervention is a validated peer-counseling intervention, which has been validated for use in Malawi and Zimbabwe and is currently being assessed for use in Hanoi, Vietnam [25,27,28]. The SHARP intervention recommended the following (though treatment decisions are ultimately the decision of the provider and each patient): PHQ-9 scores of 0–4 does do not have recommended treatment, scores of 5–9 should receive the Friendship Bench (FB) intervention, and scores of 10 or higher should consider prescription treatments [25,26].

**Study setting and participants.**  Inclusion criteria for the observational cohort were being age 18–65 years, seeking hypertension or diabetes care at a participating NCD clinic, and having depression at baseline (defined as a PHQ-9 score >4). All participants completed the PHQ-9 with a clinician during routine clinical care to determine depressive symptom eligibility for the parent study. Exclusion criteria (Fig 1) were history of bipolar or psychotic disorder or showing emergent self-harm threat. All participants gave written, informed consent in Chichewa or Chitumbuka. This consent included a consent comprehension checklist and was approved by all relevant institutional review boards (IRBs). After screening and consent, participants were re-assessed for depressive symptoms as part of the baseline interview by a research assistant within one week of the screening interview. Participants who were consented and enrolled in the parent study (n = 203) who had a PHQ-9 score ≥5 at screening, but a PHQ-9 score <5 in the baseline interview, were excluded from this analysis [25]. While these participants were eligible for participation at screening and enrollement, we do not know why their PHQ-9 score was lower during the baseline interview. We are only interested in those patients who had depressive symptoms at baseline.

The SHARP study data used for this analysis were collected between May 16, 2019 and June 27, 2022 via structured baseline, three, six, and 12-month questionnaires (with their measures and instruments described in detail below). Surveys at all data collection timepoints were identical, except for certain demographic variables (e.g., sex and education level) only measured during the baseline interview. Interviews were conducted in private settings in participants' clinic sites before March 25, 2020 and via telephone from March 25, 2020 onwards due to the SARS-CoV-2 pandemic. Participants were excluded from respective analyses if they had not completed a given follow-up interview (n = 40, 61, and 126 at three, six, and 12 months respectively) or if they were missing exposure, outcome, or covariate data (n = 6, Fig 1).

### Measures

**Depression.**  Each interview included the 9-item PHQ-9 measuring depressive symptoms in the last 2 weeks [29–32]. First, we collected data on each of the nine depressive symptoms as a continuous score [29,30]. Next, we summed each participant's number of reported

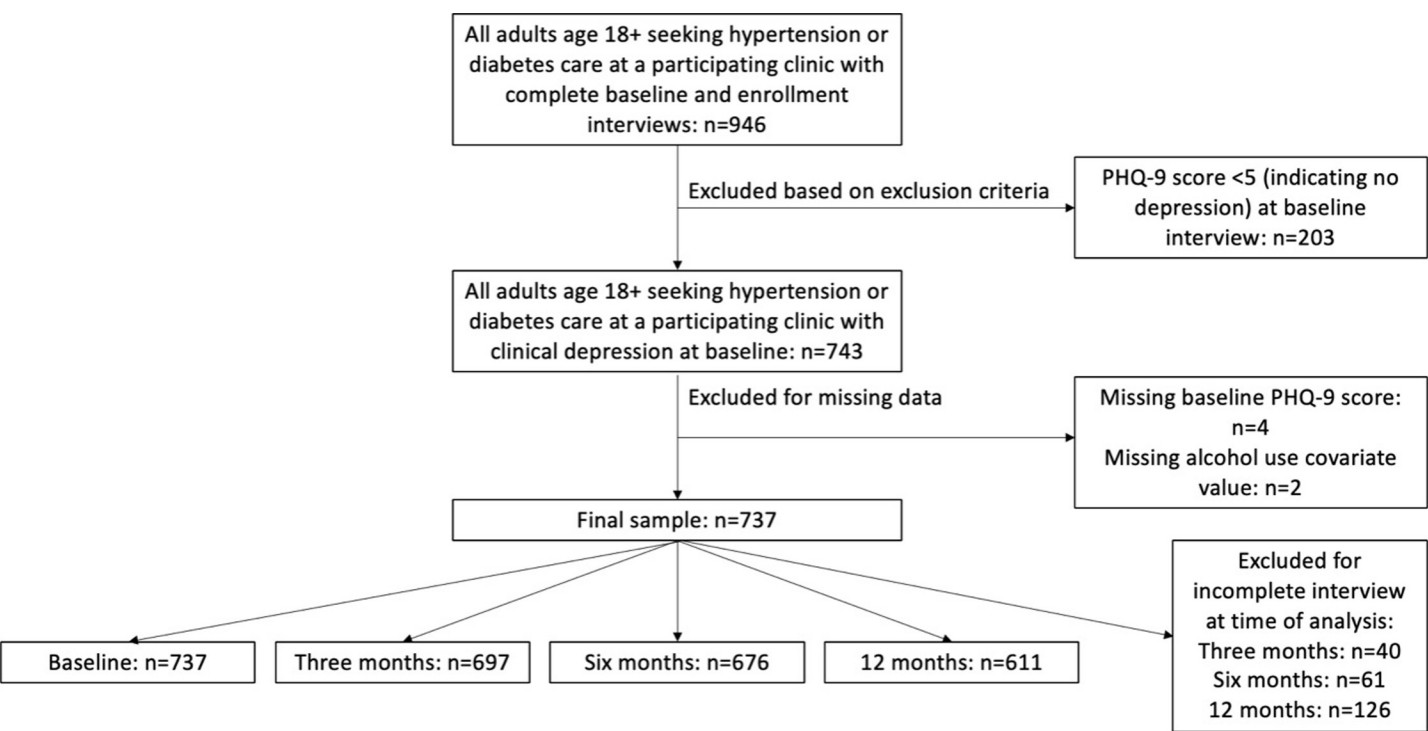

**Fig 1. Participant flow chart: Inclusion and exclusion criteria.**

depressive symptoms to measure overall depressive severity at each follow-up time point, with no, mild, moderate, moderately severe, and severe depression indicated by PHQ-9 scores of 0–4, 5–9, 10–14, 15–19, and 20–27, respectively [25,29,32]. Finally, we dichotomized PHQ-9 score at each follow-up time point to assess this study's primary outcome of interest: remission (PHQ-9 score ≤4) vs. no remission (PHQ-9 score ≥5) [25,29,32].

**SLEs.** The SLE questionnaire was adapted from the Life Experiences Survey, validated for assessment of 57 stressful life experiences in adults [33–35]. The survey measured recent SLEs included experiencing death or illness of a friend or family member; hospitalization; a major new health problem; experiencing assault, robbery, and/or a motor vehicle accident; financial stress (measured by number of days going to bed hungry in the last month); employment changes and challenges; and relationship changes (e.g., marriage, divorce, and estrangement) [33,34,36,37]. SLEs not prevalent in this study population, but which were on the original LES survey, were excluded (e.g., student related stressors). Participants were asked a series of questions (SF 2) to determine presence or absence of each of these SLEs in the three months prior to each interview (e.g., at baseline, participants were asked about SLEs in the three months prior to the baseline interview; at the three month follow-up interview, participants were asked about SLEs in the three months prior to this interview (after the baseline interview and before the three month follow-up). Reported SLEs were summed and used as a continuous exposure variable in the final model, which was determined as the best functional form based on Akaike information criterion (AIC) estimates and visualization on locally weighted scatterplot smoothing (LOWESS) plots [35,38,39].

**Covariates.** Possible confounders, mediators, and modifiers were identified through directed acyclic graphs (DAGs) and substantive knowledge (Fig 2). Covariates included in the final adjustment set were identified through DAGs (created based on current literature and

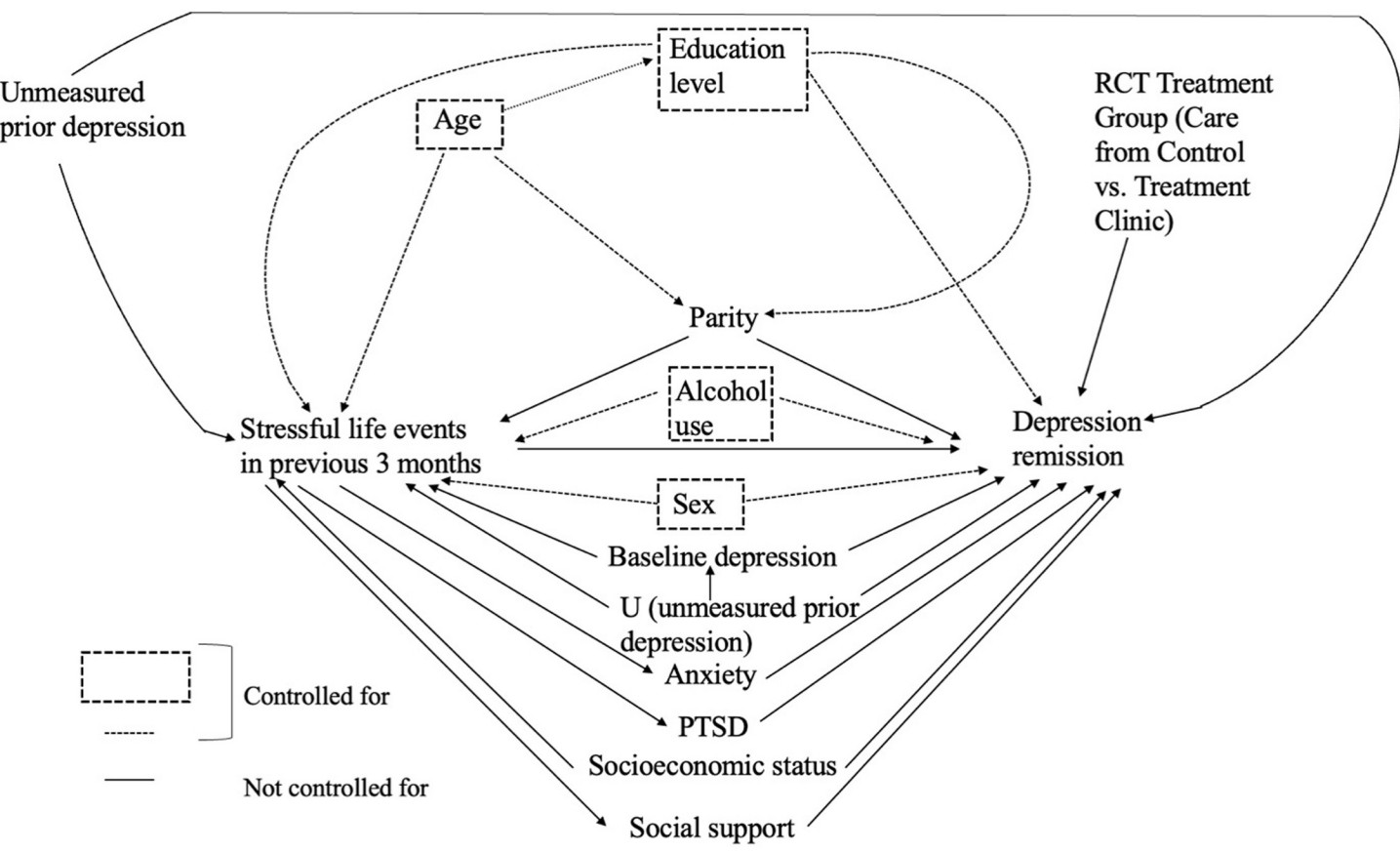

**Fig 2. Directed Acyclic Graph (DAG): Stressful life events and depression remission.**

subject matter knowledge), with likelihood ratios for nested models and AIC estimates for nested and non-nested models informing categorization choice in the final regression model. Mediation analysis was beyond the scope of this study.

The final adjustment set (and functional forms) included sex (binary: female, male), education level (ordinal: none, primary grades 1–5, primary grade 6–8, secondary, postsecondary), and alcohol use (ordinal: never, monthly or less, 2-4x per month, 2-3x per week, ≥4x per week). Study arm and clinic site were included as fixed effects, not as confounders, in the final model. Inclusion of a wealth index score as a proxy for socioeconomic status (SES) did not change effect estimates or model precision (suggesting no confounding by SES in the study sample) and resulted in convergence issues in one of the models. As such, SES was not included as a covariate in the final models. GAD-7 scores were not included in final models, as anxiety is a likely mediator between SLEs and depression remission achievement, but is reported for participants at baseline (Table 1) [40,41].

## Statistical analysis

This study measures depression remission incidence at three, six, and 12 months of follow-up. We used a log-binomial form of the generalized linear model (GLM) to obtain parameter estimates (cumulative incidence ratios [CIR]), assessing exposure effect per additional SLE on probability of achieving depression remission. We also report the CIR per 3 (median) additional SLEs on depression remission probability. We use Poisson risk regression with robust

**Table 1. SHARP participant characteristics at baseline.**

| | | N (%) | Mean | Median | SD | Min | Max |
|---|---|---|---|---|---|---|---|
| | | 737 | | | | | |
| Gender | Female | 580 (78.7) | | | | | |
| | Male | 157 (21.3) | | | | | |
| | Missing | 0 (0.0) | | | | | |
| Age (range: 18–65) | | | 50.8 | 52 | 9.9 | 18 | 65 |
| | 18–29 | 22 (3.0) | | | | | |
| | 30–39 | 83 (11.3) | | | | | |
| | 40–49 | 196 (26.6) | | | | | |
| | 50+ | 436 (59.2) | | | | | |
| | Missing | 0 (0.0) | | | | | |
| Education | | | | | | | |
| | No formal schooling | 117 (15.9) | | | | | |
| | Standard 1–5 | 237 (32.2) | | | | | |
| | Standard 6–8 | 219 (29.7) | | | | | |
| | Secondary school | 127 (17.2) | | | | | |
| | Post-secondary school | 37 (5.0) | | | | | |
| | Missing | 0 (0.0) | | | | | |
| Marital status | | | | | | | |
| | Never married | 13 (1.8) | | | | | |
| | Currently married | 488 (66.1) | | | | | |
| | Separated | 68 (9.3) | | | | | |
| | Divorced | 44 (6.0) | | | | | |
| | Widowed | 121 (16.4) | | | | | |
| | Cohabiting with a partner | 3 (0.4) | | | | | |
| | Missing | 0 (0.0) | | | | | |
| Parity[1] | | | 5.8 | 6.0 | 2.8 | 0 | 14 |
| | 0 children (nulliparity) | 16 (2.8) | | | | | |
| | 1–4 children (low multiparity) | 175 (30.2) | | | | | |
| | 5–8 children (grand multiparity) | 292 (50.3) | | | | | |
| | 9–14 children (great grand multiparity) | 95 (16.4) | | | | | |
| | Missing | 2 (0.3) | | | | | |
| Study arm[2] | | | | | | | |
| | Y | 393 (53.3) | | | | | |
| | Z | 344 (46.7) | | | | | |
| Work status | | | | | | | |
| | Employed | 699 (94.8) | | | | | |
| | Unemployed | 34 (4.6) | | | | | |
| | Missing | 4 (0.5) | | | | | |
| Alcohol use | Ever | 28 (3.8) | | | | | |
| | Never | 709 (96.2) | | | | | |
| Depression severity at baseline | Mild depressive symptoms | 476 (64.6) | 9.3 | 8 | 3.8 | 5 | 25 |
| | Moderate depressive symptoms | 187 (25.4) | | | | | |
| | Moderately severe depressive symptoms | 56 (7.6) | | | | | |
| | Severe depressive symptoms | 18 (2.4) | | | | | |
| Stressful life events (SLEs) at baseline | ≥0, <2 | 156 (21.2) | 3.5 | 3 | 2.4 | 0 | 15 |
| | ≥2, <3 | 151 (20.5) | | | | | |
| | ≥3, <5 | 215 (29.2) | | | | | |
| | ≥5, <15 | 215 (29.2) | | | | | |

[1]Among female participants.

[2]Parent study arm, investigators are blinded to intervention arm status.

standard errors (SEs) to estimate CIRs due to convergence issues when restricting observation periods [42,43]. The linearity assumption for the exposure variable was supported by LOWESS plot visualization and lower AIC for the linear model compared to the quadratic, categorical (quartiles and quintiles), top-coded quadratic models, and restricted cubic spline models. When comparing the linear and quadratic functional forms, the quadratic beta coefficient was non-statistically significant, further supporting the linearity assumption.

## SARS-CoV-2 pandemic

Importantly, the severe acute respiratory syndrome coronavirus 2 (SARS-CoV-2) pandemic, beginning partway through this study in December 2019, may have influenced the frequency of SLEs and severity of depression among participants, as SLEs related to illness, death, and hospitalization due to coronavirus disease 19 (COVID-19) have increased and availability of depression care has decreased globally during the pandemic [3,44–47]. All participants had depression at baseline, so incidence of depression pre- and during the pandemic is not able to be measured.

We defined the pre-pandemic period as May 16, 2019-March 24, 2020 and the during-pandemic period as March 25, 2020 to present. March 25, 2020 was the date that in-person SHARP follow-up was shifted to telephone-based follow-up. No study participants had been enrolled long enough to complete 12-month interviews in the pre-pandemic period. SARS-CoV-2 descriptive analyses were restricted to nine of 10 study clinics, as one clinic had completed enrollment before the pandemic period.

We explored four questions in order to understand potential impacts of the SARS-CoV-2 pandemic on this paper's primary analyses: 1) Did the distribution of covariates and participant characteristics differ among those enrolled before and during the pandemic?, 2) Did the distribution of the exposure (number of SLEs) differ between the pre- and during pandemic periods?, 3) Did the distribution of the outcome (depression remission) change before and during the pandemic, 4) Are there changes in enrollment rates overall and at each study site before and during the pandemic in a way that might inform interpretation of results? Finally, we assessed the association of SLEs at baseline and probability of depression remission at each time point in both time periods. We used a Poisson model with robust SEs to estimate the CIR and Wald test for modification as the log-binomial model failed to converge in these analyses [42,43].

## Ethical approval

The National Health Sciences Research Committee of Malawi (NHSRC; Approval # 17–3110) and the University of North Carolina at Chapel Hill Biomedical Institutional Review Board (Approval #17/11/1925) approved all aspects of this research.

## Results

### Study sample

SHARP enrolled n = 960 participants between May 16, 2019 and December 13, 2021 who all had at least mild depressive symptoms at clinical screening, n = 946 of whom completed the baseline survey and continued in the study. Of those who completed baseline surveys, n = 743 participants (Table 1) had at least mild depressive symptoms after enrollment at the baseline research interview and were eligible for inclusion in this analysis. At the time of this analysis, n = 40, n = 61, and n = 126 participants were excluded for incomplete interview status at three, six, and 12 months, respectively (Fig 1). Participants were excluded for missing baseline PHQ-

9 data (n = 4) and alcohol use (n = 2). No participants were missing data on the exposure, education, or sex.

The study sample consisted of primarily females (n = 580, 79%). Most participants were 50 years of age or older (n = 436, 59%, mean: 50.8, standard deviation (SD): 9.9). Nearly two thirds of participants had primary education level from grades 1–8 (n = 456, 62%). The mean number of children reported by female participants was 5.8 children (SD: 2.8; range: [0,14]) and half of women reported having 5–8 children (n = 292, 50%). Nearly the entire study sample reported employment at baseline (n = 699, 95%). Few participants reported ever using alcohol at baseline (n = 28, 3.8%).

## SLEs and depression remission

Overall, average depressive severity decreased and incidence of depression remission increased over follow-up (Table 2). Approximately 40%, 50%, and 54% of participants had achieved depression remission at three, six, and 12 months, respectively. Mild depressive severity was the most common severity level among those still living with depression at three, six, and 12 months.

When estimating the association of SLEs with depression remission, unadjusted and adjusted results were very similar (Table 3). In the adjusted model (Fig 3A), each additional SLE was associated with a 6% relative decrease in the probability of having depression remission at three months among participants eligible to start treatment (CIR: 0.94; 95% CI [0.90, 0.98]; Confidence Limit Ratio (CLR): 1.09; p = 0.002). The estimated, adjusted CIR at three months is compatible with a range of true CIRs from 0.90 to 0.98 (α = 0.05).

At six months, each additional SLE was associated with a 5% relative decrease in the probability of having depression remission (Fig 3B) (0.95; [0.92,0.98]; CLR: 1.07; p = 0.002). The estimated, adjusted CIR at six months is compatible with a range of true CIRs from 0.92 to 0.98 (α = 0.05). Similarly, at 12 months each additional SLE was associated with a 4% relative decrease in the probability of having depression remission (Fig 3C) (0.96; [0.94, 0.99], CLR: 1.06; p = 0.011). The estimated, adjusted CIR at 12 months is compatible with a range of values 0.94 to 0.99.

Given that the median number of SLEs experienced in the sample at baseline was 3 SLEs, we re-expressed regression results as the change in probability of depression remission associated with an increase of 3 SLEs. The adjusted probability of achieving depression remission at

**Table 2. PHQ-9 scores at baseline, three months, six months, and 12 months.**

|  |  | 3 months | 6 months | 12 months |
|---|---|---|---|---|
|  | N | 697 | 676 | 611 |
| **Depression remission** | Yes | 278 (39.9) | 336 (49.7) | 330 (54.0) |
|  | No | 419 (60.1) | 340 (50.3) | 281 (46.0) |
| **Total depression score** | Mean (SD, range) | 5.8 (3.9, [0,22]) | 5.1 (3.9, [0,22]) | 4.7 (3.8, [0,23]) |
|  | Median | 5 | 5 | 4 |
| **Depressive severity** | No depressive symptoms (depression remission) | 278 (39.9) | 336 (49.7) | 330 (54.0) |
|  | Mild depressive symptoms | 322 (46.2) | 258 (38.2) | 231 (37.8) |
|  | Moderate depressive symptoms | 75 (10.8) | 65 (9.6) | 36 (5.9) |
|  | Moderately severe depressive symptoms | 18 (2.6) | 12 (1.8) | 9 (1.5) |
|  | Severe depressive symptoms | 4 (0.6) | 5 (0.7) | 5 (0.8) |

[1]Follow-up interviews not completed at the time of analysis were considered missing and excluded from analysis.

**Table 3. Unadjusted and adjusted cumulative incidence ratio (CIR) estimates.**

| Follow-up | Unadjusted CIR (95% CI) | p (α = 0.05) | Adjusted CIR (95% CI) | p (α = 0.05) |
|---|---|---|---|---|
| **For each additional SLE** | | | | |
| 3-months | 0.93 (0.89, 0.97) | 0.001 | 0.94 (0.90, 0.98) | 0.002 |
| 6-months | 0.95 (0.92, 0.99) | 0.005 | 0.95 (0.92, 0.98) | 0.002 |
| 12-months | 0.96 (0.93, 0.99) | 0.022 | 0.96 (0.94, 0.99) | 0.011 |
| **For 3 additional SLEs (median exposure in sample)** | | | | |
| 3-months | 0.80 (0.70, 0.91) | 0.001 | 0.82 (0.72, 0.93) | 0.002 |
| 6-months | 0.86 (0.78, 0.96) | 0.005 | 0.86 (0.78, 0.95) | 0.002 |
| 12-months | 0.89 (0.81, 0.98) | 0.022 | 0.89 (0.82, 0.97) | 0.011 |

[1]The adjustment set included the following covariates: Sex (binary: Female, male), education level (categorical: None, primary grades 1–5, primary grades 6–8, secondary or postsecondary), age (categorical: 18–44 years, 45–51 years, 52+ years) and alcohol use (binary: Ever, never).

three, six, and 12 months was 0.82 [95% CI: 0.72, 0.93], 0.86 [0.78, 0.95], and 0.89 [0.82, 0.97] times lower per 3 SLEs.

## SARS-CoV-2 pandemic

Importantly, this study was conducted before and during the SARS-CoV-2 pandemic. Enrollment was higher in the pre-pandemic period compared to the pandemic period (Table 4), with the mean number of participants enrolled per month being 45.5 and 11.6 in the pre-pandemic and pandemic periods, respectively. Most clinics (80%) enrolled over half of their total enrolled participants in the pre-pandemic period. Participant characteristics (age, sex, education, alcohol use, and parity) were similar across periods. The mean number of SLEs reported at baseline among participants enrolled in the pre-pandemic period was 3.8 SLEs (SD: 2.6) while the mean number of SLEs reported by participants enrolled during the pandemic was 3.2 SLEs (SD: 2.2).

Several SLEs increased in frequency during the pandemic period (Table 4), including death of a partner, child, mother, father, sibling, or grandparent; serious illness of a sibling or grandparent; major problems at work; major new health problems oneself; motor vehicle accidents (MVAs); and being robbed. Mean PHQ-9 and Generalized Anxiety Disorder (GAD-7) scores were similar, but slightly lower in the during-pandemic compared to pre-pandemic periods. Overall, the CIR at three and six months were higher in the during pandemic period compared to the pre-pandemic period, although statistical significance varied (Table 4).

## Discussion

In this sample of participants with diabetes and/or hypertension who were identified as eligible for depression treatment, SLEs were common, with most participants (94%) reporting at least one SLE and 79% reporting more than one SLE. Among those who reported at least mild depression symptoms at baseline, full remission was achieved by 40%, 50%, and 54% of participants at three, six, and 12 months, respectively. Mild depression was the most frequent depressive severity among those with depression at all time points, which is consistent with prior literature assessing depression severity in primarily adult women in Sub-Saharan Africa [22,48,49].

Recent SLEs were associated with lower risk of achieving depression remission at three, six, and 12 months in this sample. These findings suggest that SLEs are risk factors for changes in

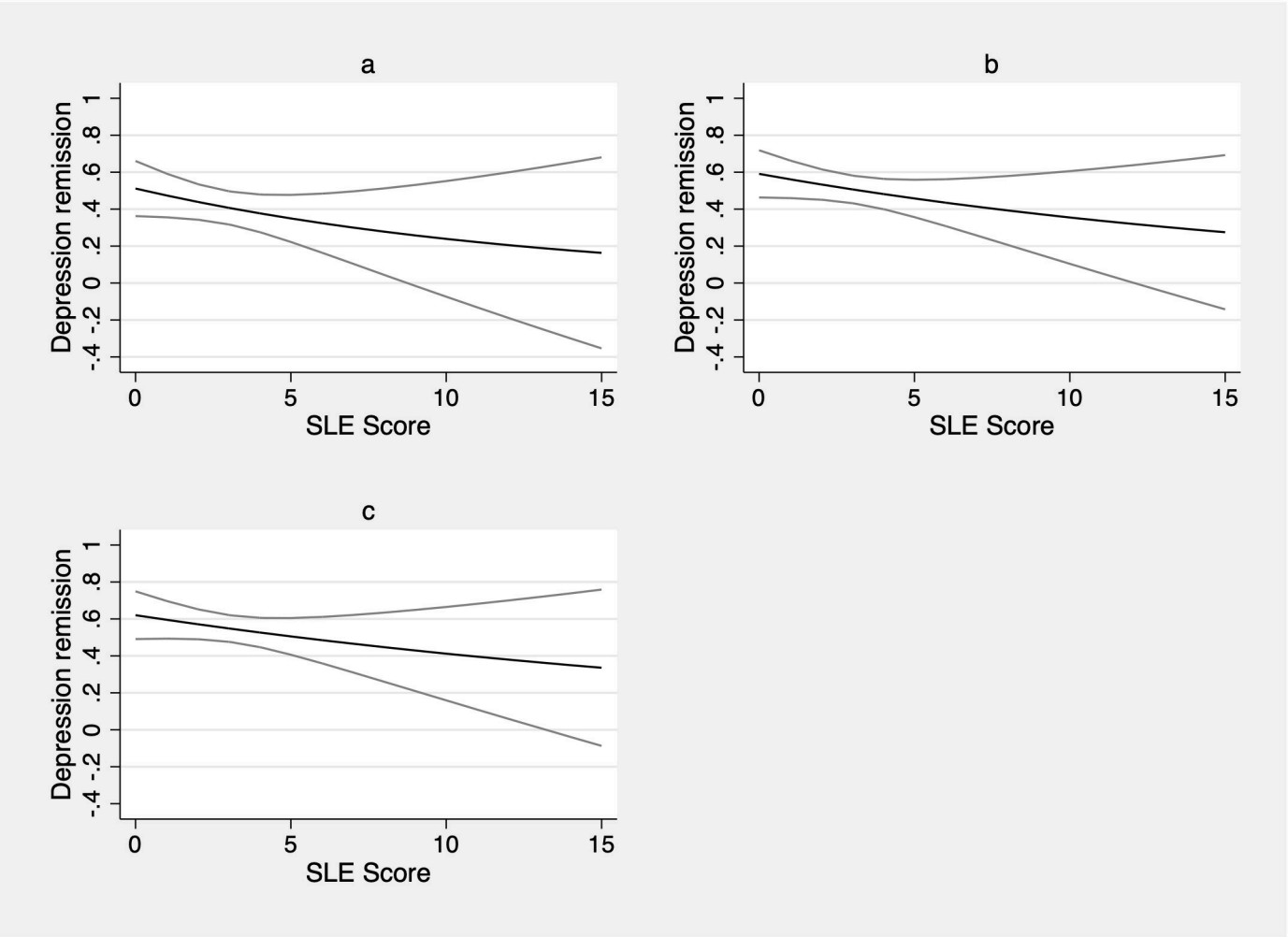

**Fig 3. Baseline stressful live events and depression remission over time: Adjusted cumulative incidence ratios (CIRs).** a. Baseline stressful life events and 3-month depression remission: Adjusted CIR b. Baseline stressful life events and 6-month depression remission: Adjusted CIR c. Baseline stressful life events and 12-month depression remission: Adjusted CIR.

depressive severity, a finding consistent with the current SLE and depression literature, and specifically for probability of depression remission achievement among patients indicated for depression treatment in integrated NCD-depression care models [50–54]. Further, these results are consistent with prior studies suggesting that the impact of SLEs on depression remission may decrease over time [55,56]. These findings signal opportunity to help identify individuals at risk of severe depression and TRD and possible intervention points for these individuals [20]. Depression diagnoses are often missed in NCD clinic settings in Malawi, and knowledge of recent SLEs may help identify patients at risk of depression, particularly TRD [57].

Importantly, this study was conducted before and during the SARS-CoV-2 pandemic during which SLE frequency and severity may have been higher during the study period than would have been anticipated had there been no pandemic. The proportion of participants reporting a death of a partner, child, sibling, or grandparent, serious illness of a mother, sibling, or grandparent, major problems at work, a major new health problem, MVAs, and experiencing robbery increased in the during-pandemic period compared to the pre-pandemic period. Increases in death and health events are plausible due to infection and fatality due to

**Table 4. Pre-pandemic and during pandemic frequency distributions during baseline interviews.**

|  | Pre-pandemic | During pandemic |  |
|---|---|---|---|
| **Mean number of participants enrolled each month** | 45.5 | 11.6 | |
| **Participant characteristics (%)** | | | |
| **Age (mean, SD)** | 50.9 (SD: 9.6) | 50.5 (SD: 10.5) | |
| **Education (%)** | | | |
| No formal schooling | 57 (14.8) | 39 (14.8) | |
| Standard grades 1–5 | 125 (32.6) | 70 (26.6) | |
| Standard grades 6–8 | 118 (30.7) | 84 (31.9) | |
| Secondary school | 64 (16.7) | 53 (20.2) | |
| Postsecondary school | 20 (5.2) | 17 (6.5) | |
| **Sex** | | | |
| Male | 79 (20.6) | 65 (24.7) | |
| Female | 305 (79.4) | 198 (75.3) | |
| **Parity (mean, SD)[1]** | 6.1 (2.7) | 5.5 (2.8) | |
| **Alcohol use (any; %)** | 17 (4.4) | 10 (3.8) | |
| **SLEs at baseline (mean, SD)** | 3.8 (SD: 2.6) | 3.2 (SD: 2.2) | |
| Death of a partner (%) | 4 (1.0) | 7 (2.7) | |
| Death of a child (%) | 14 (3.7) | 11 (4.2) | |
| Death of a mother (%) | 11 (2.9) | 8 (3.0) | |
| Death of a father (%) | 8 (2.1) | 6 (2.3) | |
| Death of a sibling (%) | 32 (8.3) | 22 (8.4) | |
| Death of a grandparent (%) | 7 (1.8) | 10 (3.8) | |
| Serious illness of a sibling (%) | 31 (8.1) | 29 (11.0) | |
| Serious illness of a grandparent (%) | 3 (0.8) | 4 (1.5) | |
| Major problems at work (%) | 22 (5.7) | 21 (8.0) | |
| Major new health problem oneself (%) | 114 (29.7) | 91 (34.6) | |
| Motor vehicle accident | 6 (1.6) | 8 (3.0) | |
| Robbed | 60 (15.6) | 43 (16.4) | |
| **PHQ9 score (mean, SD)** | 9.9 (SD: 4.1) | 8.8 (SD: 3.5) | |
| **GAD7 score (mean, SD)** | 7.7 (SD: 4.3) | 6.5 (SD: 3.9) | |
|  | **CIR [(95% CI), p][2]** | **CIR [(95% CI), p][2]** | **Wald Test [(95% CI), p][3]** |
| Three months | 0.90 [(0.84, 0.96), 0.002] | 1.03 [(0.96, 1.11), 0.436] | [(1.01, 1.12), 0.031] |
| Six months | 0.91 [(0.86, 0.96), 0.001] | 0.98 [(0.93, 1.03), 0.456] | [(0.95, 1.05), 0.901] |
| 12 months | - | 0.95 [(0.93, 0.99), 0.012] | - |

[1]among female participants.

[2]Using the Poisson form of GLMs with robust standard errors.

[3] [43].

SARS-CoV-2 infection and COVID-19 disease, as well as potential worsening of existing health conditions due to more difficult access to care or SARS-CoV-2 infection complications [44,58–61]. All participants had depression at study baseline, precluding measurement of new, incident depression cases. Interestingly, there was not a higher mean PHQ-9 score or GAD-7 score at baseline in the pandemic period compared to the pre-pandemic period in the study sample [62–64]. This finding may be due to factors unrelated to the pandemic. Alternatively, depression and anxiety might have increased in the target population overall during the pandemic, with those with more severe depression and anxiety preferentially avoiding care due to

the pandemic and those who were observed having lower depression and anxiety levels [3,62,65–67].

## Study strengths and limitations

Study strengths include validity of outcome variable ascertainment, as the PHQ-9 was validated for use in this context, and consistency in collection of exposure and outcome measures, with trained research assistants collecting all information at all time points. Additionally, this study makes use of a large cohort of patients recruited from 10 geographically diverse facilities providing NCD care in Malawi and followed for 12 months after indication for depression treatment.

Unmeasured depression history prior to study baseline may confound the effect estimates, as it may affect baseline, three-month, six-month, and 12-month probability of depression remission and exposure to new SLEs in the three months prior to baseline (e.g., relationship and employment challenges and stressors) [68]. However, despite lifetime depression history being unmeasured, inclusion of baseline depression in models did not change effect estimates or model precision suggesting no confounding by prior depression in this study sample given that unmeasured prior depression is blocked by baseline depression on the causal pathway. Further, we measured anxiety and post-traumatic stress disorder (PTSD) symptoms at baseline. Inclusion of anxiety and PTSD scores in our model also did not change effect estimates or model precision. This suggests lack of confounding of the SLE to depression remission pathway by anxiety and stress disorder status at baseline, which was expected given these are mediators on the SLE to depression remission pathway and we would not have included either score in our models for this reason (Fig 2). Mental health can be highly stigmatized, which may bias the effect estimate towards the null if symptoms are underreported. Confounders may also be measured with error (e.g., alcohol use may be underreported among participants due to stigma). Finally, selection bias may be present, as individuals who are unable to seek care due to confounding covariates, who represent the most severe cases of depression, and for whom SLEs decrease access to care may not be represented in the study population, possibly biasing the effect estimate toward the null [69,70]. All participants were screened for depression in this study, and the results may be relevant to those adults accessing hypertension and/ or diabetes care who also have depressive symptoms in low-resource settings in Sub-Saharan Africa.

## Conclusion

Integration of depression and NCD treatment is a critical step to reducing the mental healthcare gap in low-resource settings. SLEs were predictors of depression remission over 12 months of follow-up and may represent possible intervention points and identifying factors for individuals at risk of severe depression and TRD. Assessment of SLEs during routine care is a potential method of mitigating their impact on depressive severity and improving treatment outcomes, including depression remission. Based on the study findings, our recommendations for research and practice are to consider assessment and discussion of SLEs in integrated adult NCD and depression care and treatment interventions in low-resource settings.

## Supporting information

**S1 File. PHQ-9 Survey.**
(PDF)

**S2 File. Stressful life events survey.**
(PDF)

**S3 File. Inclusivity in global research.**
(PDF)

## Acknowledgments

We greatly appreciate the time and efforts of all research staff and study participants.

## Author Contributions

**Conceptualization:** Bradley N. Gaynes, Mina C. Hosseinipour, Jones Masiye, Brian W. Pence.

**Data curation:** Bradley N. Gaynes, Harriet Akello, Jullita Kenala Malava, Josée M. Dussault, Mina C. Hosseinipour, Michael Udedi, Jones Masiye, Chifundo C. Zimba, Brian W. Pence.

**Formal analysis:** Kelsey R. Landrum, Brian W. Pence.

**Funding acquisition:** Kelsey R. Landrum, Mina C. Hosseinipour, Jones Masiye, Brian W. Pence.

**Investigation:** Kelsey R. Landrum, Bradley N. Gaynes, Harriet Akello, Jullita Kenala Malava, Josée M. Dussault, Mina C. Hosseinipour, Michael Udedi, Jones Masiye, Chifundo C. Zimba, Brian W. Pence.

**Methodology:** Kelsey R. Landrum, Brian W. Pence.

**Project administration:** Kelsey R. Landrum, Bradley N. Gaynes, Mina C. Hosseinipour, Chifundo C. Zimba, Brian W. Pence.

**Supervision:** Bradley N. Gaynes, Brian W. Pence.

**Visualization:** Kelsey R. Landrum.

**Writing – original draft:** Kelsey R. Landrum.

**Writing – review & editing:** Kelsey R. Landrum, Bradley N. Gaynes, Harriet Akello, Jullita Kenala Malava, Josée M. Dussault, Mina C. Hosseinipour, Michael Udedi, Jones Masiye, Chifundo C. Zimba, Brian W. Pence.

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
