## [Decision Letter · Decision Letter 0]

17 Jul 2023

PONE-D-23-01412The longitudinal association of stressful life events with depression remission among patients with depression and hypertension or diabetes in MalawiPLOS ONE

Dear Dr. Landrum,

Thank you for submitting your manuscript to PLOS ONE. After careful consideration, we feel that it has merit but does not fully meet PLOS ONE’s publication criteria as it currently stands. Therefore, we invite you to submit a revised version of the manuscript that addresses the points raised during the review process.

We look forward to receiving your revised manuscript.

Kind regards,

Sathish Rajaa

Academic Editor

PLOS ONE

Journal Requirements:

"This study received funding from the National Institute of Mental Health (MPIs: Brian W. Pence, PhD, Mina C. Hosseinipour, MD MPH; Jones Masiye; U19MH113202-01). KRL received support from supplement U19MH113202-04S1. JMD received support from supplement U19MH113202-01S2 and institutional training grant T32AI070114. No funding agency or funder had any part in study conceptualization, data collection, data analysis, manuscript preparation, or publication decision-making."     

Additional Editor Comments:

Thank you for the opportunity to serve as the Editor for the manuscript titled "The longitudinal association of stressful life events with depression Remission among patients with depression and Hypertension or diabetes in Malawi". The author's work for the manuscript is commendable. However, the manuscript looks statistically sound but methodologically weak. I have a lots of major concerns that need to be addressed to make this manuscript an important one for the readers of this journal and consider acceptance

General considerations:

Kindly spell check the entire manuscript for grammatical errors and use of punctuation

Title:  Consider adding the study design to the title

 Abstract:

Please stick to the journal guidelines for abstract preparation

Introduction:

1. Introduction's first paragraph is highly confusing. It is, however, not clear what estimates the authors are providing in the first paragraph. On depression or NCDs. Be specific in providing the estimates - using a funnel approach - burden globally, in LMICs and then in Malawi - for NCD, SLEs, depression, its association and the possible effect of SLE on remission

2. Too many references quoted in the first two paragraphs, there are almost 20 references quoted in the first 2 paragraphs. This opens up a question is so much literature is available why do a primary study again?

3. Please mention the rationale and novelty of the study and what the study results are expected to add

Methods and Results:

1. Please elaborate more on the study setting, explaining the mental health and NCD care context that exists in Malawi. Service provision and the care pathway that exists for the treatment of a patient with NCD or depression

2. Did the authors obtain the necessary permission for the use of data from the SHARP study

3. The parent study is also authored by the same authors? Please declare the COI - if any for the use of data from the previous study

4. How did the authors obtain a cohort from an RCT? The RCT has a controlled environment? can a cohort mimicking natural history be taken from an RCT?

5. Please explain more on the acquisition of data

6. What is exactly the study design? is it a cohort? or a plain observational analysis?

7. This line "Those participants (n=203) who had a PHQ-9 score ≥5 at screening and consent but a PHQ-9 score <5 in the baseline interview were excluded from this analysis" is highly misleading and contradicts the inclusion criteria. Please be specific and clear

8. Please elaborate more on the SLE questionnaire, if it's a validated instrument.? and how did the authors arrive at the specified categorisation?

9. Did the authors attempt to adjust for the possible mediators using the mediation analysis?

10. SLEs 3 moths prior to the interviews were linked with remission of depression recorded during the time of the interview? Was the sufficient theoretically possible time for the development of depression following SLEs considered before associating SLE and depression remission?

11. In case the patients reported SLE a day prior to interview and is found to have depression right now and no depression during follow-ups.. how was such a scenario handled in the manuscript?

12. Why SLE score of 2 taken as separate category?

13. There is no mention o Table 5 in the manuscript

14. Was the effect of COVID-19 on the incidence of depression and the SLE adjusted in the analysis?

Discussion:

The discussion looks very weak and poorly written. It is not enough when we plainly mention similar studies, rather efforts need to be made to compare and contrast the results and findings across the study settings. Please rewrite the discussion part discussing more on: i) comparison with previous studies ii) biological or theoretical reasons for the estimates obtained iii) Clinical implications iv) Programmatic or policy implications

Many references are not in Vancouver format

Reviewers' comments:

Reviewer's Responses to Questions

**Comments to the Author**

1. Is the manuscript technically sound, and do the data support the conclusions?

Reviewer #1: Yes

Reviewer #2: Yes

2. Has the statistical analysis been performed appropriately and rigorously? 

Reviewer #1: I Don't Know

Reviewer #2: Yes

3. Have the authors made all data underlying the findings in their manuscript fully available?

Reviewer #1: Yes

Reviewer #2: Yes

4. Is the manuscript presented in an intelligible fashion and written in standard English?

Reviewer #1: Yes

Reviewer #2: Yes

5. Review Comments to the Author

Reviewer #1: The authors have selected an interesting topics. The conclusion is supported by the results. Addressing stressful life events during integrated non communicable disease and depression care interventions can improve success of depression treatment among adult patients.

Minor comnents:- you have used old references eg reference 25,50,51,52

Reviewer #2: Hello. Thank you for the opportunity to review this interesting article. I think it was an valuable endeavor and done quite well in the face of the pandemic and other barriers.

I understand the conclusion suggests that recent SLEs affect time to remission for depression. Although, that does sounds reasonable , there are a few questions I had in this regard-

1) What was the treatment modality employed to treat the depression? Was there any variation between the different groups in terms of what depression modality was used which could confound the results. You can mention what the treatment was in the methods.

2) Was there a difference in the hypertension treatment/ actually blood pressure control of the two studied groups. That could also present a variable that might confound.

3) The final conclusion you came to was the screening for SLEs might help in mitigating impact on depression severity and improving treatment- How? Again, we don't know what was used to treat the patients so we can't comment on how knowing SLEs might affect the treatment or the treatment choices. They might affect expectations of remission for the clinicians. You could elaborate on this in your conclusion or in your strengths/ limitations. I think depression screening would definitely be helpful but have to elaborate on the helpfulness of SLE screening.

6. PLOS authors have the option to publish the peer review history of their article (what does this mean?). If published, this will include your full peer review and any attached files.

Reviewer #1: **Yes: **Agmas Wassie Abate

Reviewer #2: No

---

## [Author Response · Author response to Decision Letter 0]

2 Nov 2023

Dear PLOS ONE Editors and Reviewers,

Thank you so much for your time and responses. We greatly appreciate your thoughtful questions and comments and we hope that these changes address all of your concerns. Please find our responses in line below and the corresponding revisions in the revised manuscript.

"This study received funding from the National Institute of Mental Health (MPIs: Brian W. Pence, PhD, Mina C. Hosseinipour, MD MPH; Jones Masiye; U19MH113202-01). KRL received support from supplement U19MH113202-04S1. JMD received support from supplement U19MH113202-01S2 and institutional training grant T32AI070114. No funding agency or funder had any part in study conceptualization, data collection, data analysis, manuscript preparation, or publication decision-making." 

 "This study received funding from the National Institute of Mental Health (MPIs: Brian W. Pence, PhD, Mina C. Hosseinipour, MD MPH; Jones Masiye; U19MH113202-01). KRL received support from supplement U19MH113202-04S1. JMD received support from supplement U19MH113202-01S2 and institutional training grant T32AI070114. The funders had no role in study design, data collection and analysis, decision to publish, or preparation of the manuscript." 

Additional Editor Comments:

Thank you for the opportunity to serve as the Editor for the manuscript titled "The longitudinal association of stressful life events with depression Remission among patients with depression and Hypertension or diabetes in Malawi". The author's work for the manuscript is commendable. However, the manuscript looks statistically sound but methodologically weak. I have a lots of major concerns that need to be addressed to make this manuscript an important one for the readers of this journal and consider acceptance

General considerations:

Kindly spell check the entire manuscript for grammatical errors and use of punctuation

We have further proofread the manuscript for technical, grammatical, and punctuation errors. Thank you.

Title:

Consider adding the study design to the title

The title, “The longitudinal association of stressful life events with depression remission among SHARP trial participants with depression and hypertension or diabetes in Malawi”, describes that this is a longitudinal analysis. We have added “among a cohort” to clearly show that this is analysis is situated in the SHARP RCT. 

Abstract:

Please stick to the journal guidelines for abstract preparation

Introduction:

1. Introduction's first paragraph is highly confusing. It is, however, not clear what estimates the authors are providing in the first paragraph. On depression or NCDs. Be specific in providing the estimates - using a funnel approach - burden globally, in LMICs and then in Malawi - for NCD, SLEs, depression, its association and the possible effect of SLE on remission

Thank you for this comment. We have edited the introduction accordingly. We agree that this approach makes it clearer to the reader that this is a study of the longitudinal association between depression and SLEs among patients that have depression (who also happen to have hypertension and/or diabetes, though NCDs are not the focus of the study themselves.) 

2. Too many references quoted in the first two paragraphs, there are almost 20 references quoted in the first 2 paragraphs. This opens up a question is so much literature is available why do a primary study again?

Thank you for this comment. After rearranging and editing the introduction, we hope that this addresses this comment. We conducted a thorough review of the literature (global, nationally, and sub-nationally) 

3. Please mention the rationale and novelty of the study and what the study results are expected to add

We have extended this section of the introduction, thank you for this comment.

Methods and Results:

1. Please elaborate more on the study setting, explaining the mental health and NCD care context that exists in Malawi. Service provision and the care pathway that exists for the treatment of a patient with NCD or depression

We have elaborated on the context of mental health care service provision, including where the SHARP trial is situated in integrated mental health and NCD service provision in the country. We have added this to expand the introduction, and further expand the introduction according to the above comments.

2. Did the authors obtain the necessary permission for the use of data from the SHARP study

Yes, the authors are SHARP study team members. 

3. The parent study is also authored by the same authors? Please declare the COI - if any for the use of data from the previous study

There are a multitude of published manuscripts from the SHARP trial, with some of the same coauthors as this manuscript. There are not conflicts of interest, as declared by all authors during manuscript submission. Funding sources and agencies had no role in study conceptualization, data collection, data analyses, manuscript preparation or reporting, or any other study processes.

4. How did the authors obtain a cohort from an RCT? The RCT has a controlled environment? can a cohort mimicking natural history be taken from an RCT?

All data are drawn from the prospective RCT. After accounting for study site and arm in regression analyses, this allows us to look at a cohort of RCT patients prospectively and longitudinally. We have added more discussion regarding this. Thank you for bringing up this point. We hope that our clarifications address this concern.

5. Please explain more on the acquisition of data

As also described above, we have elaborated on data ascertainment in the Methods section. The study instruments are described in the Measures section of the Methods section. Given that we are the research team for the SHARP study, we collect, manage, and report the data ourselves.

6. What is exactly the study design? is it a cohort? or a plain observational analysis?

7. This line "Those participants (n=203) who had a PHQ-9 score ≥5 at screening and consent but a PHQ-9 score <5 in the baseline interview were excluded from this analysis" is highly misleading and contradicts the inclusion criteria. Please be specific and clear

Thank you for this comment. We have clarified this in this manuscript section. 

“Participants who were consented and enrolled in the parent study (n=203) who had a PHQ-9 score ≥5 at screening, but a PHQ-9 score <5 in the baseline interview, were excluded from this analysis.1 While at screening, these participants were eligible for participation, we do not know why their PHQ-9 score, which should be relatively stable over a short period of time, was lower during the baseline interview.”

8. Please elaborate more on the SLE questionnaire, if it's a validated instrument.? and how did the authors arrive at the specified categorisation?

Yes, the LES is a validated instrument for assessment of 57 SLEs in adults.2–4 The instrument was adapted to address categories of SLEs specific to this study population: experiencing death or illness of a friend or family member; hospitalization; a major new health problem; experiencing assault, robbery, and/or a motor vehicle accident; financial stress (measured by number of days going to bed hungry in the last month); employment changes and challenges; and relationship changes (e.g., marriage, divorce, and estrangement).2,3,5,6 The original LES survey includes SLEs that are not relevant in this context and that were excluded from our survey: foreclosure on mortgage or loan and student related SLES (e.g., failing an important exam, failing a course, dropping a course). 2–4 

Thank you for this comment. We have provided additional information and rationale for decision making.

9. Did the authors attempt to adjust for the possible mediators using the mediation analysis?

We did not conduct a mediation analysis. Anxiety and PTSD may be mediators between SLEs and depression remission, and including adjusting for these variables in our current would result in biased estimates. When included in the primary analysis model (e.g., at the 3mo time point), inclusion of anxiety as a confounder changes the effect estimate substantially (∼10% increase in the adjusted CIR) while inclusion of ptsd does not change the effect estimate substantially (<5% increase in the adjusted CIR). A formal mediation analysis is beyond the scope of this paper, but there is evidence for anxiety and ptsd being associated with stressful life events and depression (or lack of depression remission in our case, as they often occur comorbidly).7–13

10. SLEs 3 moths prior to the interviews were linked with remission of depression recorded during the time of the interview? Was the sufficient theoretically possible time for the development of depression following SLEs considered before associating SLE and depression remission?

Thank you for this comment. At each interview, participants were asked about SLEs experience in the three months BEFORE/LEADING UP TO that interview. So, at 3 months, participants were asked about SLES in the three months prior to that interview (the three months after the baseline interview and before the 3 month follow up interview). We have ensured clarification of this in the manuscript and hope that this addresses these questions.

11. In case the patients reported SLE a day prior to interview and is found to have depression right now and no depression during follow-ups.. how was such a scenario handled in the manuscript?

This is an interesting question, thank you for raising this point. It is possible that the participant experienced a SLE 1 day before the interview, but screens negative for depression. This participant would be considered as not having depression at that interview. The impact of SLEs can vary across and within individuals over time, so it is possible they experienced an SLE but are not experiencing depression symptoms at that time point.14–18 

12. Why SLE score of 2 taken as separate category?

The SLE measure is a continuous measure. As such, we report the mean (SD), median, minimum, and maximum values for participants at baseline. We have also reported SLEs as a categorical variable, based on quartiles of the continuous SLE distribution (≥0, <2; ≥2, <3; ≥3, <5; ≥5, <15)In our results (Table 3), we report change in CIR per SLE, and per 3 SLEs (the median number of SLEs experienced in the study cohort).

13. There is no mention o Table 5 in the manuscript

Thank you for this comment, this should have said Table 4 and is now corrected.

14. Was the effect of COVID-19 on the incidence of depression and the SLE adjusted in the analysis?

We cannot measure incident depression in this analysis, as all participants have depression at study baseline (therefore, there can be no new, incident cases diagnosed). Further, we could not have anticipated having to do this additional analysis during the SHARP study’s design, as we did not anticipate having to measure variables related to a pandemic. As such, we assess if the CIR of depression remission changes before and during the pandemic. We report mean PHQ-9 scores before and during the pandemic, number and proportion of each type of SLE before and during the pandemic, and CIRs before and during the pandemic. We hope this answers your question, thank you for this point.

Discussion:

The discussion looks very weak and poorly written. It is not enough when we plainly mention similar studies, rather efforts need to be made to compare and contrast the results and findings across the study settings. Please rewrite the discussion part discussing more on: i) comparison with previous studies ii) biological or theoretical reasons for the estimates obtained iii) Clinical implications iv) Programmatic or policy implications

Many references are not in Vancouver format

Thank you for these comments. We have further expanded comparison of study results to other studies and hope that this sufficiently addresses your concerns. 

Reviewers' comments:

Reviewer's Responses to Questions

Comments to the Author

1. Is the manuscript technically sound, and do the data support the conclusions?

Reviewer #1: Yes

Reviewer #2: Yes

2. Has the statistical analysis been performed appropriately and rigorously? 

Reviewer #1: I Don't Know

Reviewer #2: Yes

3. Have the authors made all data underlying the findings in their manuscript fully available?

Reviewer #1: Yes

Reviewer #2: Yes

4. Is the manuscript presented in an intelligible fashion and written in standard English?

Reviewer #1: Yes

Reviewer #2: Yes

5. Review Comments to the Author

Reviewer #1: The authors have selected an interesting topics. The conclusion is supported by the results. Addressing stressful life events during integrated non communicable disease and depression care interventions can improve success of depression treatment among adult patients.

Thank you for your comments and time in reading this manuscript. We hope that, collective with all comments and questions, we have improved the manuscript to address all concerns and strengthen the manuscript as a whole.

Minor comnents:- you have used old references eg reference 25,50,51,52

Thank you for this point. Reference 25 refers to the original PHQ-9 citation (the instrument itself). We also provide validation studies for the PHQ-9 tool for use in this context. References 50-52 is are key early studies in SLE and depression research, and cover a large scope of foundational research on this topic. We hope that, with more recent SLE and depression studies cited and added, this is sufficient to provide a review of what is currently known (and unknown) in the depression and SLE field of research.19–24

Reviewer #2: Hello. Thank you for the opportunity to review this interesting article. I think it was an valuable endeavor and done quite well in the face of the pandemic and other barriers.

I understand the conclusion suggests that recent SLEs affect time to remission for depression. Although, that does sounds reasonable , there are a few questions I had in this regard-

1) What was the treatment modality employed to treat the depression? Was there any variation between the different groups in terms of what depression modality was used which could confound the results. You can mention what the treatment was in the methods.

2) Was there a difference in the hypertension treatment/ actually blood pressure control of the two studied groups. That could also present a variable that might confound.

3) The final conclusion you came to was the screening for SLEs might help in mitigating impact on depression severity and improving treatment- How? Again, we don't know what was used to treat the patients so we can't comment on how knowing SLEs might affect the treatment or the treatment choices. They might affect expectations of remission for the clinicians. You could elaborate on this in your conclusion or in your strengths/ limitations. I think depression screening would definitely be helpful but have to elaborate on the helpfulness of SLE screening.

Thank you very much for your comments and for your time in reviewing our manuscript, and helping strengthen our reporting of this study’s results. Please find our responses in line below. We hope that they sufficiently address your questions and concerns.

1) What was the treatment modality employed to treat the depression? Was there any variation between the different groups in terms of what depression modality was used which could confound the results. You can mention what the treatment was in the methods.

Thank you for this comment. We have added discussion of this to the methods section accordingly. Clinics were randomized to one of two depression screening and treatment strategies, either a basic or enhanced implementation package. The basic implementation package involves a trained provider supporting SHARP implementation. The enhanced implementation package involves a trained provider supporting the intervention and a quality assurance program. 

Participants receive standard of care in both packages. The SHARP intervention recommended the following, though treatment decisions are the decision of the provider and each patient: PHQ-9 score 0-4 does not have recommended treatment, if the patient has a score of 5-9 it is recommended that they receive the Friendship Bench (FB) intervention, and if the patient has a PHQ-9 score of 10 or higher it is recommended that they consider prescription depression treatment options.1,25 The FB intervention is a validated peer-counseling intervention, which has been validated for use in Malawi and Zimbabwe and is currently being assessed for use in Hanoi, Vietnam.1,26,27

2) Was there a difference in the hypertension treatment/ actually blood pressure control of the two studied groups. That could also present a variable that might confound.

This is a very interesting point. Hypertension treatment was based on provider expertise and patient-provider decision making. We believe that, though hypertension and cardiovascular disease may be linked to depression 19–23, hypertension is a mediator on the pathway from SLEs to depression or depression remission. SLEs and chronic stress can lead to cardiovascular diseases and are also related to hypertension, which may impact depression and depression remission.24 

3) The final conclusion you came to was the screening for SLEs might help in mitigating impact on depression severity and improving treatment- How? Again, we don't know what was used to treat the patients so we can't comment on how knowing SLEs might affect the treatment or the treatment choices. They might affect expectations of remission for the clinicians. You could elaborate on this in your conclusion or in your strengths/ limitations. I think depression screening would definitely be helpful but have to elaborate on the helpfulness of SLE screening.

These are excellent comments and points, thank you. The rational for this statement is that depression is frequently undiagnosed in the study context, and identifying patients at particularly high risk for depressive disorders may increase detection and treatment of depression cases that otherwise would have been missed. Depression diagnoses are often missed in NCD clinic settings in Malawi, and knowledge of recent SLEs may help identify patients at risk of depression, particularly TRD.28 We hope that our expansion of this discussion is suitable to addressing your comment, thank you again for raising these points.

6. PLOS authors have the option to publish the peer review history of their article (what does this mean?). If published, this will include your full peer review and any attached files.

Do you want your identity to be public for this peer review? For information about this choice, including consent withdrawal, please see our Privacy Policy.

Reviewer #1: Yes: Agmas Wassie Abate

Reviewer #2: No

References

1. Gaynes, B. N. et al. The Sub-Saharan Africa Regional Partnership (SHARP) for Mental Health Capacity-Building Scale-Up Trial: Study Design and Protocol. Psychiatr Serv 72, 812–821 (2021).

2. Sarason, I. G., Johnson, J. H. & Siegel, J. M. Assessing the impact of life changes: Development of the Life Experiences Survey. Journal of Consulting and Clinical Psychology 46, 932–946 (1978).

3. Sarason, I. G. & Johnson, J. H. The Life Experiences Survey: Preliminary Findings. https://apps.dtic.mil/sti/citations/ADA027527 (1976).

4. Reif, S. et al. Highly Stressed: Stressful and Traumatic Experiences among individuals with HIV/AIDS in the Deep South. AIDS Care 23, 152–162 (2011).

5. Stack, R. J. & Meredith, A. The Impact of Financial Hardship on Single Parents: An Exploration of the Journey From Social Distress to Seeking Help. Journal of Family and Economic Issues 39, 233–242 (2018).

6. Guan, N., Guariglia, A., Moore, P., Xu, F. & Al-Janabi, H. Financial stress and depression in adults: A systematic review. PLoS One 17, e0264041 (2022).

7. Kendler, K. S., Hettema, J. M., Butera, F., Gardner, C. O. & Prescott, C. A. Life Event Dimensions of Loss, Humiliation, Entrapment, and Danger in the Prediction of Onsets of Major Depression and Generalized Anxiety. Arch Gen Psychiatry 60, 789–796 (2003).

8. Michl, L. C., McLaughlin, K. A., Shepherd, K. & Nolen-Hoeksema, S. Rumination as a mechanism linking stressful life events to symptoms of depression and anxiety: Longitudinal evidence in early adolescents and adults. Journal of Abnormal Psychology 122, 339–352 (2013).

9. Kim, J. I., Park, H. & Kim, J.-H. The mediation effect of PTSD, perceived job stress and resilience on the relationship between trauma exposure and the development of depression and alcohol use problems in Korean firefighters: A cross-sectional study. Journal of Affective Disorders 229, 450–455 (2018).

10. Lowe, S. R. et al. Pathways from assaultive violence to post-traumatic stress, depression, and generalized anxiety symptoms through stressful life events: longitudinal mediation models. Psychological Medicine; Cambridge 47, 2556–2566 (2017).

11. Buccheri, T., Musaad, S., Bost, K. K. & Fiese, B. H. Development and assessment of stressful life events subscales – A preliminary analysis. Journal of Affective Disorders 226, 178–187 (2018).

12. Weinberg, A., Kujawa, A. & Riesel, A. Understanding Trajectories to Anxiety and Depression: Neural Responses to Errors and Rewards as Indices of Susceptibility to Stressful Life Events. Curr Dir Psychol Sci 31, 115–123 (2022).

13. Flory, J. D. & Yehuda, R. Comorbidity between post-traumatic stress disorder and major depressive disorder: alternative explanations and treatment considerations. Dialogues Clin Neurosci 17, 141–150 (2015).

14. Hammen, C. Risk factors for depression: An autobiographical review. Annu. Rev. Clin. Psychol. 14, 1–28 (2018).

15. Kendler, K. S., Thornton, L. M. & Gardner, C. O. Stressful Life Events and Previous Episodes in the Etiology of Major Depression in Women: An Evaluation of the “Kindling” Hypothesis. AJP 157, 1243–1251 (2000).

16. Monroe, S. & Harkness, K. L. Life Stress, the ‘Kindling’ Hypothesis, and the Recurrence of Depression: Considerations From a Life Stress Perspective. Psychological Review 112, 417–445 (2005).

17. Post, R. M. Transduction of psychosocial stress into the neurobiology of recurrent affective disorder. AJP 149, 999–1010 (1992).

18. Stroud, C. B., Davila, J., Hammen, C. & Vrshek-Schallhorn, S. Severe and nonsevere events in first onsets versus recurrences of depression: Evidence for stress sensitization. Journal of Abnormal Psychology 120, 142–154 (2011).

19. Cohen, S., Murphy, M. L. M. & Prather, A. A. Ten Surprising Facts About Stressful Life Events and Disease Risk. Annu Rev Psychol 70, 577–597 (2019).

20. Cohen, S., Gianaros, P. J. & Manuck, S. B. A Stage Model of Stress and Disease. Perspect Psychol Sci 11, 456–463 (2016).

21. Dimsdale, J. E. Psychological Stress and Cardiovascular Disease. Journal of the American College of Cardiology 51, 1237–1246 (2008).

22. Steptoe, A. & Kivimäki, M. Stress and cardiovascular disease. Nature Reviews. Cardiology 9, 360–370 (2012).

23. Netterstrøm, B. et al. The Relation between Work-related Psychosocial Factors and the Development of Depression. Epidemiologic Reviews 30, 118–132 (2008).

24. Berntson, J., Patel, J. S. & Stewart, J. C. Number of recent stressful life events and incident cardiovascular disease: Moderation by lifetime depressive disorder. Journal of Psychosomatic Research 99, 149–154 (2017).

25. Akiba, C. F. et al. The Sub-Saharan Africa Regional Partnership (SHARP) for Mental Health Capacity Building: a program protocol for building implementation science and mental health research and policymaking capacity in Malawi and Tanzania. International Journal of Mental Health Systems vol. 13 1–13 https://doaj-org.libproxy.lib.unc.edu (2019).

26. Chibanda, D. et al. Effect of a Primary Care–Based Psychological Intervention on Symptoms of Common Mental Disorders in Zimbabwe: A Randomized Clinical Trial. JAMA 316, 2618–2626 (2016).

27. Stockton, M. A. et al. The impact of an integrated depression and HIV treatment program on mental health and HIV care outcomes among people newly initiating antiretroviral therapy in Malawi. PLoS One 15, e0231872 (2020).

28. Udedi, M., Pence, B. W., Stewart, R. C. & Muula, A. S. Detection and prevalence of depression among adult type 2 diabetes mellitus patients attending non-communicable diseases clinics in Lilongwe, Malawi. International Journal of Mental Health Systems (2020) doi:http://dx.doi.org/10.21203/rs.3.rs-39726/v2.

---

## [Decision Letter · Decision Letter 1]

26 Jan 2024

The longitudinal association of stressful life events with depression remission among SHARP trial participants with depression and hypertension or diabetes in Malawi

PONE-D-23-01412R1

Dear Dr. Landrum,

We’re pleased to inform you that your manuscript has been judged scientifically suitable for publication and will be formally accepted for publication once it meets all outstanding technical requirements.

Kind regards,

Marianne Clemence, Staff Editor, on behalf of,

Abdelhfeez Moshrif

Academic Editor

PLOS ONE

Additional Editor Comments (optional):

From my point of view, the authors' responses to reviewers are sufficient and the suggestion from Reviewer #2 is not a requirement

Reviewers' comments:

Reviewer's Responses to Questions

**Comments to the Author**

1. If the authors have adequately addressed your comments raised in a previous round of review and you feel that this manuscript is now acceptable for publication, you may indicate that here to bypass the “Comments to the Author” section, enter your conflict of interest statement in the “Confidential to Editor” section, and submit your "Accept" recommendation.

Reviewer #1: All comments have been addressed

Reviewer #2: All comments have been addressed

2. Is the manuscript technically sound, and do the data support the conclusions?

Reviewer #1: Yes

Reviewer #2: Yes

3. Has the statistical analysis been performed appropriately and rigorously? 

Reviewer #1: I Don't Know

Reviewer #2: Yes

4. Have the authors made all data underlying the findings in their manuscript fully available?

Reviewer #1: Yes

Reviewer #2: Yes

5. Is the manuscript presented in an intelligible fashion and written in standard English?

Reviewer #1: Yes

Reviewer #2: Yes

6. Review Comments to the Author

Reviewer #1: The Topic is very interesting and the authors have been addressed all the issues raised by the reviewer.

Reviewer #2: Thanks for answering my questions. Overall, I think the finding that depression remission rates decrease with the presence of SLEs. You should emphasize that in the conclusion. Of course depression screening is important but the important aspect of the study is the effect SLEs have on depression remission.

7. PLOS authors have the option to publish the peer review history of their article (what does this mean?). If published, this will include your full peer review and any attached files.

Reviewer #1: **Yes: **Agmas Wassie Abate

Reviewer #2: No

---

## [Editor Report · Acceptance letter]

16 Feb 2024

PONE-D-23-01412R1 

PLOS ONE

Dear Dr. Landrum, 

I'm pleased to inform you that your manuscript has been deemed suitable for publication in PLOS ONE. Congratulations! Your manuscript is now being handed over to our production team.

Kind regards, 

on behalf of

Professor Abdelhfeez Moshrif 

Academic Editor

PLOS ONE